# PLANNING TO GO OUT-OF-DISTRIBUTION IN OFFLINE-TO-ONLINE REINFORCEMENT LEARNING

## ABSTRACT

Offline pretraining with a static dataset followed by online fine-tuning (offline-to-online, or OtO) is a paradigm that is well matched to a real-world RL deployment process: in few real settings would one deploy an offline policy with no test runs and tuning. In this scenario, we aim to find the best-performing policy within a limited budget of online interactions. Previous work in the OtO setting has focused on correcting for bias introduced by the policy-constraint mechanisms of offline RL algorithms. Such constraints keep the learned policy close to the behavior policy that collected the dataset, but this unnecessarily limits policy performance if the behavior policy is far from optimal. Instead, we forgo policy constraints and frame OtO RL as an exploration problem: we must maximize the benefit of the online data-collection. We study major online RL exploration paradigms, adapting them to work well with the OtO setting. These adapted methods contribute several strong baselines. Also, we introduce an algorithm for **p**lanning **t**o **g**o **o**ut **o**f **d**istribution (PTGOOD), which targets online exploration in relatively high-reward regions of the state-action space unlikely to be visited by the behavior policy. By leveraging concepts from the Conditional Entropy Bottleneck, PTGOOD encourages data collected online to provide new information relevant to improving the final deployment policy. In that way the limited interaction budget is used effectively. We show that PTGOOD significantly improves agent returns during online fine-tuning and finds the optimal policy in as few as 10k online steps in Walker and in as few as 50k in complex control tasks like Humanoid. Also, we find that PTGOOD avoids the suboptimal policy convergence that many of our baselines exhibit in several environments.

## 1 INTRODUCTION

In real-world reinforcement learning (RL), there is great value in being able to train an agent offline with a static dataset. But fine-tuning the agent over at least a small number of agent-environment interactions is also key, given the risks in real-world deployment. This offline-to-online (OtO) scenario extends offline RL (also called batch RL (Ernst et al., 2005; Reidmiller, 2005)), which has garnered attention as a framework for learning control from datasets without online interactions Levine et al. (2020). While offline RL removes the potentially costly data-collection step of traditional RL, the resulting policy may well be suboptimal. This could occur if the offline dataset does not cover all areas of the state-action space relevant to our task or if the policy that collected the dataset was itself suboptimal. Given this risk, those deploying an RL agent in the real world would likely invest in fine-tuning the agent, at least over a small budget of online interactions.

In this study, we view OtO RL as an exploration problem. Because the agent has a limit on its environment interactions, it must choose carefully which state-action pairs to collect during online fine-tuning. This contrasts starkly with prior work in OtO RL, which has focused on correcting for bias introduced by the policy-constraint mechanisms used in existing offline RL algorithms (Beeson & Montana, 2022; Nakamoto et al., 2023; Luo et al., 2023). Such policy-constraint mechanisms are used during offline training to keep the learned policy close to the behavior policy that collected the offline dataset (e.g., the inclusion of a behavior-cloning term). While these methods can work well offline, they can cause detrimental learning instabilities during online fine-tuning, due to overly-conservative value functions (Nakamoto et al., 2023). Instead, **we do not use these**

**policy-constraint mechanisms at any point**. In doing so, we shift the problem set away from bias correction to data-collection strategy during the online fine-tuning phase.

While exploration is widely studied in the online RL literature, the OtO problem differs from the standard online learning setup in two unique ways. First, the OtO setting greatly constrains the number of online data-collection steps. Second, the online phase in OtO RL benefits from information available from offline pretraining. We wish to leverage the offline dataset and pretraining phase to optimize a deployment policy using a limited number of agent-environment interactions. Given exploration methods have not generally featured in the OtO RL literature, we evaluate the compatibility of major online RL exploration paradigms with the OtO setting. In particular, we analyze intrinsic motivation and upper confidence bound (UCB) exploration. We find that intrinsic-motivation methods can unlearn initializations from offline pretraining, and note that the implementation details of UCB-style methods can affect exploration behavior. Ultimately, we address the issues with these methods leading to several strong baselines for exploration in OtO RL.

From the exploration perspective, the most basic OtO question is: *Which experiences should the agent collect during online fine-tuning such that its returns improve the most in the fewest agent-environment interactions?* To address this question, we propose an algorithm for **p**lanning **t**o **g**o **o**ut **of** **d**istribution (PTGOOD) that can be exploited by any existing model-based RL algorithm. PTGOOD uses a learned density of state-action pairs in the offline dataset to collect transitions during online fine-tuning that are out-of-distribution relative to the data in the offline dataset. By targeting such state-action pairs, PTGOOD continually increases the diversity of the information available in the total (offline plus online) data. PTGOOD also targets high-reward state-action pairs by ensuring that exploration guidance does not stray too far from the current-best policy, to ensure *relevance* of the collected data. We also note that PTGOOD uses the learned density in a non-myopic planning procedure, thereby considering exploration fruitfulness in future steps.

Our experiments demonstrate that PTGOOD consistently and significantly outperforms other exploration baselines in terms of evaluation returns and avoids suboptimal policy convergence, a problem we find with many exploration methods in several environments. In addition, we find that PTGOOD often finds the optimal policy in simpler environments such as Walker in as few as 10k online steps and in as few as 50k in more complex control tasks like Humanoid. Our contributions can be summarized as follows:

- We propose PTGOOD, a non-myopic planning algorithm for OtO exploration that targets high-reward out-of-distribution transitions via an estimate of the regions of the state-action space represented in the offline dataset.
- We systematically study online RL exploration methods, identify compatibility issues with the OtO setting, and propose well-performing baselines that overcome those issues.
- We collect, benchmark, and open-source new offline datasets in addition to the usual D4RL (Fu et al., 2020) datasets, and evaluate PTGOOD and other baselines on these.

## 2 BACKGROUND

The RL problem usually studies an agent acting within a Markov decision process (MDP) parameterized by the tuple $(\mathcal{S}, \mathcal{A}, \mathcal{T}, R, \gamma)$. $\mathcal{S}, \mathcal{A}$ are the state- and action-spaces, respectively, $\mathcal{T}(s'|s, a)$ is the transition function that describes the distribution over next-states conditioned on the current state and action, $R(s, a)$ is the reward function, and $\gamma \in (0, 1)$ is the discount factor. The agent acts within the MDP according to its policy $\pi(a|s)$, which maps states to a distribution over actions. An agent's policy $\pi$ induces a (discounted) occupancy measure $\rho_\pi(s, a)$, which is the stationary distribution over the $\mathcal{S} \times \mathcal{A}$ space unique to policy $\pi$ (Syed et al., 2008; Kang et al., 2018). After executing an action $a_t$ in state $s_t$ at timestep $t$, the next state is sampled $s_{t+1} \sim \mathcal{T}(\cdot|s_t, a_t)$, the agent receives a reward $r_t = R(s_t, a_t)$, and the interaction loop continues. The agent's learning objective is to find a policy that maximizes cumulative discounted returns $\pi^* = \arg\max_\pi \mathbb{E}_\pi[\sum_{t=1}^{\infty} \gamma^{t-1} R(s_t, a_t)]$. Model-based RL approaches learn a model of the MDP's transition function $\hat{\mathcal{T}}$ and reward function $\hat{R}$, which can then be used to generate rollouts of "imagined" trajectories from a given state $s_t$: $\tau = (s_t, a_t, \hat{r}_t, \hat{s}_{t+1}, \dots)$.

OtO RL assumes access to a dataset of transition tuples $D_{\pi_b} = \{(s, a, r, s')_i\}_{i=1}^{|D_{\pi_b}|}$ collected by some (potentially) unknown behavior policy $\pi_b$. In addition, the behavior policy's performance can

range from that of a random agent to an expert agent, which means that $D_{\pi_b}$ may contain trajectories of highly-suboptimal behavior. The goal in OtO RL is to leverage offline data $D_{\pi_b}$ to determine a policy $\pi_o$ to collect another dataset $D_{\pi_o}$ over a fixed-budget of agent-environment interactions, which are then used together $D_{\pi_b} \cup D_{\pi_o}$ to try to find optimal policy $\pi^*$. We need to optimize over both the choice of final policy and the data collection process that leads to that final policy.

## 3 RELATED WORK

**Exploration in RL.** Exploration is a key problem in RL and has been studied extensively in the online setting. Exploration algorithms cover many strategies such as dithering methods like $\epsilon$-greedy or randomized value functions (Osband et al., 2016). Intrinsic reward methods leverage prediction error (Pathak et al., 2017; Burda et al., 2019) and count-based rewards (Andoni & Indyk, 2008; Ostrovski et al., 2017) to guide agents towards unseen regions of the state-action space. Upper confidence bound (UCB) methods use uncertainty to guide agent exploration. For example, some algorithms measure uncertainty as disagreement within ensembles of Q-functions (Chen et al., 2017; Lee et al., 2021a; Schäfer et al., 2023) or transition functions (Shyam et al., 2019; Henaff, 2019; Sekar et al., 2020). In contrast to these methods, PTGOOD uses prior information explicitly by estimating a density of already-collected data and uses this density to plan exploration.

**Offline RL.** Many offline RL methods are designed to constrain the learned policy to be similar to the behavior policy. For example, conservative methods incorporate their policy constraint either via behavior cloning terms (Wu et al., 2019; Peng et al., 2019; Fujimoto & Gu, 2021), restricting the policy-search space (Kumar et al., 2021), restricting the policy's action space (Fujimoto et al., 2019), or incorporating policy-divergence regularization into the critic (Nachum et al., 2019; Kostrikov et al., 2021). On the other hand, pessimistic methods suppress the value of out-of-distribution state-action pairs, disincentivizing the agent from traversing those regions. For example, Kidambi et al. (2020) and Yu et al. (2020) penalize value based on disagreement between transition models, Rigter et al. (2022) use an adversarial world model to generate pessimistic transitions, and (Kumar et al., 2020) penalize the value of actions too different from ones the behavior policy would choose.

**OtO RL.** Some research in the OtO RL setting involves empirical studies of algorithm implementation choices. For example, Lee et al. (2021b) develop a replay sampling mechanism to mitigate large errors in bootstrap value function updates, and Ball et al. (2023) study choices like using LayerNorm and sampling proportions between offline and online data. Most previous work in the OtO setting targets over-conservatism induced by a given offline RL algorithm (Beeson & Montana, 2022; Nakamoto et al., 2023; Luo et al., 2023). In contrast, PTGOOD approaches the OtO RL setting as an exploration problem. Li et al. (2023a) show theoretically that the exploration perspective is useful for OtO in tabular MDPs when combined with pessimism. In contrast, we focus on continuous MDPs, and PTGOOD does not use conservatism or pessimism in any form.

**Control with Expert Demonstrations.** Closely related to OtO RL is learning from demonstration (LFD) (Schaal, 1996). Many LFD methods use a form of behavior cloning on expert or hand-crafted trajectories for policy initialization followed by online fine-tuning with RL operators (Hester et al., 2018; Vecerik et al., 2017; Rajeswaran et al., 2018). In contrast, we study a setting where the learned policy has **no** prior access to demonstrations from expert or hand-crafted policies.

## 4 PLANNING TO GO OUT OF DISTRIBUTION

Later, in §5, we look at existing intrinsic reward and upper confidence bound (UCB) exploration methods, and adapt them appropriately for the OtO setting. However, these exploration methods are lacking in two respects: (a) UCB methods are myopic and rely on ensemble-based uncertainty to drive exploration, and (b) intrinsic reward methods use a moving-target reward function which can cause instabilities in value function training which translates to instabilities in policy training. This leads us to develop PTGOOD, an approach that overcomes these issues. We introduce PTGOOD first before looking at other comparator exploration methods.

PTGOOD drives exploration by estimating $\rho_{\pi_b}$, the occupancy measure for policy $\pi_b$ (effectively the marginal density over states and actions), and then planning over a tree of imagined rollouts to collect state-action pairs within low-density regions. PTGOOD considers not just a single state-

action pair but selects state-action pairs for data collection that are within a few transitions (in terms of $\mathcal{T}(\cdot)$) of many other low-likelihood state-action pairs. We posit that data collected during online fine-tuning in the OtO setting should meet two criteria: (1) be non-redundant to data in the offline dataset and (2) be of relatively high reward.

PTGOOD satisfies criterion (1) through the use of the Conditional Entropy Bottleneck (CEB) (Fischer, 2020) to model the density of state-action pairs. PTGOOD satisfies criterion (2) by ensuring that the exploration guidance does not stray too far from the improving policy. This is accomplished by sampling the policy and adding a small amount of noise during the planning process. As the policy updates target high-reward regions in the vicinity of the current policy, exploring "close" to the policy is important. The notion of closeness is explored later in §6.4.

## 4.1 PTGOOD

PTGOOD can be applied as a complement to any model-based offline RL method. Given a learnt offline policy and dynamics model, PTGOOD plans the data collection process a step at a time to collect the next transition, which then augments the offline data and all data collected so far. The policy can now be updated with the new data. The data-collection planning process can then be repeated as many times as our budget of online interactions allows.

---

**Algorithm 1** PTGOOD Planning Procedure

---

**Input:** Dynamics model $\hat{\mathcal{T}}$, encoder $e$, marginal $m$, depth $d$, width $w$, state $s$, policy $\pi$, noise hyperparameter $\epsilon$
 1: Sample $\pi$ with state $s$, add sampled noise, and repeat $w$ times
 2: Forward-step prediction with $\hat{\mathcal{T}}$ for each sampled action and store new current states
 3: **for** $i$ in $d$ **do**
 4:     Sample $\pi$ with new current states, add sampled noise, and repeat $w$ times
 5:     Measure rate $\mathcal{R}$ for each new current state and sampled action
 6:     **for** each new current state **do**
 7:         Compute and store the expected $\mathcal{R}$ across all $w$ sampled actions
 8:     **end for**
 9:     Forward-step prediction with $\hat{\mathcal{T}}$ for each sampled action and store new current states
10: **end for**
11: Sum the stored expected $\mathcal{R}$ (Step 7) back up the chain of predicted forward steps (Steps 2 & 9) to the original $w$ sampled actions (Step 1)
**Output:** $a$ from first $w$ sampled actions (Step 1) with the highest $\mathcal{R}$ sum (Step 11)

---

The planning part of this process is given in Algorithm 1. PTGOOD's planning procedure has a width $w$ and a depth $d$. Starting from a given state $s$, we sample the policy $w$ times and add a small amount of randomly-sampled Gaussian noise $\mathcal{N}(0, \epsilon)$ with variance hyperparameter $\epsilon$ to the actions. Then, the learned dynamics model $\hat{\mathcal{T}}$ predicts one step forward from state $s$ for each $w$ actions, and action sampling is repeated with each new state. The sampling and forward-step process is repeated $d$ times, forming a tree of possible paths from the original state $s$. A key part of this algorithm is the subsequent scoring of state-action pairs using the *rate* $\mathcal{R}$, which is described in the next section.

## 4.2 THE RATE $\mathcal{R}$ AND MODELING $\rho_{\pi_b}$

The *rate* is used to measure how out-of-distribution a sample is. Rate has been used successfully in computer vision as a thresholding tool for out-of-distribution detection and has been shown to work well with CEB representations that we use here (Fischer, 2020). To ensure non-redundancy, we wish to collect low-probability state-action pairs according to $\rho_{\pi_b}$. While, in general, we do not have access to $\rho_{\pi_b}$, the offline dataset is filled with its samples. Hence we can model the offline data and use that model to target samples from occupancy measures of policies other than $\pi_b$.

We fit, to convergence, an encoder $e(z_X|x)$ and backward encoder $b(z_{X'}|x')$ to a latent space $Z$, via a standard CEB objective (described further in Appendix D.1, and given Equation 4). We use state-action pairs sampled uniformly at random from the offline dataset for $x$ and use multiplicative noise drawn from a uniform distribution $u \sim U(0.99, 1.01)$ to form $x' = u \odot x$. Next, we learn a

marginal $m(z_X)$ of our training data in the representation space of the encoder $e(\cdot)$ as a mixture of Gaussians. See Appendix D for more details. Given this encoder conditional density $e$, and marginal $m$, the *rate* (Alemi et al., 2018a;b) of a given state-action pair $x$ is computed as:

$$\mathcal{R}(x) \triangleq \log e(z_X|x) - \log m(z_X). \tag{1}$$

## 5 ADAPTING ONLINE EXPLORATION METHODS TO THE OtO SETTING

One should question whether a new algorithm such as PTGOOD is really necessary. Aren't existing exploration methods sufficient? In this section, we explore this question and then show experimentally in the following section that PTGOOD overcomes the disadvantages of existing methods.

Motivated by the lack of current OtO exploration algorithms, we now examine intrinsic reward (§5.1) and UCB exploration (§5.2) methods in the OtO setting. We find that offline initializations can be destroyed when the intrinsic rewards introduced during online fine-tuning are too large relative to the true rewards used during offline pretraining. On the other hand, if intrinsic rewards are too small, the guided exploration yields no benefit. We suggest using two agents via the DeRL framework (Schäfer et al., 2022). Here, both agents are pretrained offline, and one receives only the true rewards while the other receives intrinsic rewards during online fine-tuning. Also, with UCB methods, we find that the choice of ensemble over which uncertainty is computed changes exploration behavior. Despite the popularity of Q-function ensembles, it is not clear whether collecting data to reduce value uncertainty is better than reducing uncertainty in other learned components, such as transition functions in model-based algorithms. Ultimately, we examine the performance of using different ensembles to drive UCB exploration in our main experiments (§6.3).

### 5.1 INTRINSIC REWARDS

Intrinsic-reward methods guide exploration through a reward function that gives a bonus reward for relatively unexplored areas of the state space, the action space, or both. For example, Random Network Distillation (RND) (Burda et al., 2019) trains a network to predict the output of a fixed randomly-initialized network that transforms an incoming state. Here, the prediction error is used as a reward bonus. In this case, prediction error should be relatively high in unseen states, thereby leading the agent to explore unseen areas of the state space. Exploration is impossible during offline pretraining, which means that intrinsic rewards can only accomplish guided exploration during online fine-tuning. This leaves us to use stage-dependent reward functions: one for exploitation during offline pretraining and one for exploration during online fine-tuning.

We hypothesize that the relative magnitudes between the two rewards during online fine-tuning can complicate using intrinsic rewards in the OtO setting. For example, consider a situation where we use the modified reward at timestep $t$ as the sum of the MDP's true (extrinsic) reward $r_t^e$ and a weighted intrinsic reward $r_t^i$: $r_t = r_t^e + \lambda r_t^i$. If the intrinsic reward is too small relative to the extrinsic reward, we risk the exploration guidance having little-to-no influence on action selection. On the other hand, if the intrinsic reward is too large relative to the extrinsic reward, we risk destroying the initialization of the pretrained critic, which destroys the initialization of the pretrained actor.

To test our hypothesis, we evaluate RND agents with $\lambda \in \{0, 0.1, 1, 10, 50\}$ in two environment-dataset combinations. Specifically, we use the Halfcheetah (Random) dataset from D4RL (Fu et al., 2020) and collect our own dataset from the DeepMind Control Suite (Tassa et al., 2018; 2020) in the Walker environment, which we call DMC Walker (Random). Both datasets were collected with behavior policies that select actions uniformly at random.[1] All agents are pretrained offline with the true rewards, fine-tuned online over 50k agent-environment interactions with RND intrinsic rewards, and use Model-Based Policy Optimization (MBPO) (Janner et al., 2019) combined with Soft Actor-Critc (SAC) (Haarnoja et al., 2017) as the base agent.[2] Every 1k environment steps, we collect the agents' average undiscounted returns over ten evaluation episodes.

Figure 1 reports the average (bold) $\pm$ one standard deviation (shaded area) across five seeds. We note that when $\lambda$ is relatively small in Halfcheetah (Random), the agents perform roughly the same

---

[1]For more details on environments and datasets, see Appendix C.

[2]For more details on agents, see Appendix D

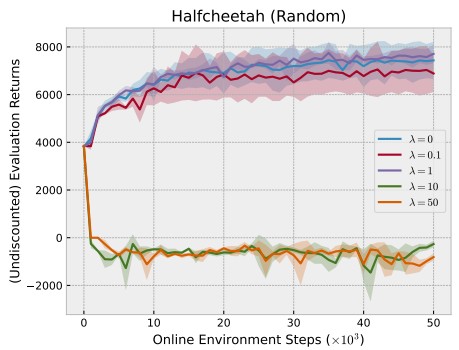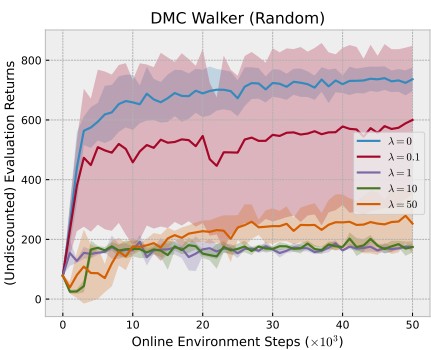

Figure 1: Undiscounted evaluation returns in Halfcheetah (Random) (left) and DMC Walker (Random) (right) for $\lambda \in \{0, 0.1, 1, 10, 50\}$ intrinsic-reward weights throughout online fine-tuning.

as when no exploration guidance is used (i.e., $\lambda = 0$). In contrast, a relatively large $\lambda$ causes the agents to lose their pretrained initialization, as shown by the dramatic drop in evaluation returns at the beginning of online fine-tuning. Our hypothesis is also confirmed in DMC Walker (Random), with the added phenomenon of bi-modal returns across seeds occurring when $\lambda = 0.1$.

We propose using two agents to overcome this issue: one for exploitation and one for exploration. Such a framework has been shown to improve learning stability in Decoupled RL (DeRL) (Schäfer et al., 2022). Both agents can be initialized with offline pretraining, but the exploitation agent only receives the MDP's true rewards, while the exploration agent receives the modified rewards during online fine-tuning. We only care about the exploitation agent for evaluation purposes and rely on the exploration agent for data collection. We refer to this agent as RND/DeRL.

## 5.2 Upper Confidence Bound Exploration

UCB-style algorithms (Auer, 2002) direct exploration on the principle of "optimism in the face of uncertainty". Many recent implementations of this principle use ensembles of Q-functions to select actions $a_t$ at timestep $t$ according to a mixture of value and uncertainty: $a_t = \arg\max_a Q_{\text{mean}}(s_t, a) + \lambda Q_{\text{std}}(s_t, a)$ (e.g., Lee et al. (2021a); Schäfer et al. (2023)). Despite reward (and therefore value) being an important component in RL, it is unclear whether it is better to follow value uncertainty or the uncertainty in another learned component.

In general, model-based RL algorithms have four core learned components that are trained with different prediction targets, learning dynamics, or both. For example, MBPO+SAC trains transition and reward functions via standard supervised learning, value functions with Bellman backups and bootstrapped targets, and policies with value and entropy maximization. Given the aforementioned differences, can we reasonably expect the uncertainty of each component to drive exploration into the same regions of the state-action space?

To answer this question, we first train an MBPO+SAC agent with ensembles of all four previously-mentioned components on the Halfcheetah (Random) dataset and evaluate their uncertainties on 2,500 transition tuples from the Halfcheetah (Expert) dataset. We evaluate the ensembles' uncertainty on a dataset collected by an expert behavior policy, as it is likely to contain out-of-distribution tuples relative to the random dataset, which is where we ultimately care about evaluating uncertainty in the OtO setting. We repeat this exercise with datasets from the Hopper environment from D4RL. If uncertainty is the same across all learned components, we should expect to see a strong positive rank correlation between each pair of ensembles' uncertainty over the expert tuples. Table 1 shows Spearman's rho between the learned components. We color cells in green when $\rho \geq 0.4$ and in red when $\rho \leq -0.4$ for ease of reading.

We highlight that the rank correlation varies greatly. In some cases, two ensembles agree strongly (e.g., Value and Transition in Halfcheetah); in others, they disagree strongly (e.g., Value and Policy

Table 1: Pair-wise rank correlation (Spearman's Rho) between different ensembles' uncertainty in Halfcheetah (left) and Hopper (right). We color cells in green when $\rho \geq 0.4$ and in red when $\rho \leq -0.4$ for ease of reading.

| | Reward | Value | Transition | Policy | | Reward | Value | Transition | Policy |
|---|---|---|---|---|---|---|---|---|---|
| Reward | | -0.26 | 0.20 | 0.15 | Reward | | -0.13 | 0.54 | 0.33 |
| Value | | | 0.55 | -0.41 | Value | | | -0.57 | -0.67 |
| Transition | | | | 0.08 | Transition | | | | 0.53 |
| Policy | | | | | Policy | | | | |

in Hopper) or show no relation (e.g., Transition and Policy in Halfcheetah). There is not necessarily a pattern that holds between the two environments. Hence, swapping learned components into the UCB action-selection equation would likely not result in similar data-collection behavior.

Perhaps, then, the ideal UCB-style algorithm would balance the uncertainty of each learned component. This balancing act is difficult because the range of each function contributing to the uncertainty computation may be significantly different, directly affecting the magnitude of the ensemble-disagreement quantity in the UCB equation. For example, a given environment's reward function may be bound to $[0, 1]$, while its action space is bound to $[-1, 1]$, and its state space is unbounded. Instead of devising a complex and adaptive balancing scheme in this work, we examine the effects of using different ensembles to drive exploration. Specifically, we evaluate one baseline that uses value-driven UCB (UCB(Q)) and one that uses dynamics-driven UCB (UCB(T)).

## 6 EXPERIMENTS

In our experiments, we aim to answer the following questions: (1) Can PTGOOD improve agent evaluation returns within the given agent-environment interaction budget? (2) How important is guided exploration to agent evaluation returns during online fine-tuning? (3) Are the policy-constraint mechanisms that are important in the purely-offline setting important in the OtO setting?

### 6.1 BASELINES

We carefully design baselines that reflect prominent categories of exploration strategies in RL. Also, we tune each of our baselines on a per-environment per-dataset basis and report results for the best-performing hyperparameters for each method. See Appendix A for more details and results. Unless otherwise noted, all algorithms use MBPO+SAC as the core model-based RL algorithm.

First, we use a baseline we call **RND/DeRL** that combines RND-based intrinsic rewards with two agents via the DeRL framework described in §5.1. We train the RND predictor using the offline dataset before online fine-tuning begins and periodically update the predictor's weights throughout the fine-tuning process. Second, we use baselines we call **UCB(Q)** and **UCB(T)**. The former uses the uncertainty from an ensemble of Q-functions, and the latter from an ensemble of transition functions in the UCB action-selection equation described in §5.2. Third, we use a **Naive** agent that does not differentiate between offline and online training and does not use any exploration guidance but instead samples the agent to choose actions. The Naive agent contextualizes the added benefit of guided exploration. Fourth, we evaluate **Cal-QL** (Nakamoto et al., 2023), a model-free algorithm designed specifically for the OtO setting that is built on top of CQL (Kumar et al., 2020), a pessimistic offline RL algorithm. Cal-QL was designed to correct for instabilities during online fine-tuning induced by CQL's policy constraint. We also benchmark **PROTO** (Li et al., 2023b) and **PEX** (Zhang et al., 2023), model-free methods designed for the OtO setting. PROTO uses a trust-region update on top of EQL (Xu et al., 2023) and TD3, and PEX learns a set of policies for action selection on top of IQL (Kostrikov et al., 2022). Finally, we contextualize the benefit of offline pretraining with **Scratch**, an agent that is only trained online but still has access to the offline dataset. None of the agents except for Cal-QL, PROTO, and PEX use conservatism or pessimism of any form during any stage of training. See Appendix D for architecture and hyperparameter details along with full implementation details for PTGOOD.

Table 2: Average ± one standard deviation of undiscoutned evaluation returns after 50k environment steps of online fine-tuning. Highest returns per environment-dataset combination bolded. Statistical significance is shown with blue highlight. (R)=Random and (MR)=Medium Replay.

| Algorithm | Halfcheetah (R) | DMC Walker (R) | Hopper (R) | Ant (R) | DMC Walker (MR) | Ant (MR) | Humanoid (MR) |
|---|---|---|---|---|---|---|---|
| PTGOOD | **8867 ± 88** | **897 ± 53** | **3246 ± 123** | **5624 ± 235** | **953 ± 6** | **5866 ± 114** | **15050 ± 878** |
| Naive | 7434 ± 782 | 736 ± 40 | 1576 ± 880 | 4663 ± 626 | 732 ± 21 | 4973 ± 337 | 11706 ± 3403 |
| RND/DeRL | 6782 ± 2013 | 677 ± 63 | 1818 ± 786 | 5258 ± 191 | 700 ± 164 | 4836 ± 695 | 1953 ± 1199 |
| Scratch | 7248 ± 814 | 668 ± 88 | 1231 ± 648 | 3702 ± 901 | 778 ± 93 | 4777 ± 1085 | 10723 ± 3903 |
| UCB(Q) | 7300 ± 861 | 740 ± 50 | 2037 ± 382 | 5290 ± 272 | 783 ± 75 | 5328 ± 224 | 13183 ± 885 |
| UCB(T) | 8170 ± 513 | 811 ± 68 | 2251 ± 830 | 5022 ± 299 | 772 ± 93 | 4508 ± 1364 | 12079 ± 2461 |
| Cal-QL | -317 ± 122 | 45 ± 4 | 57 ± 39 | -310 ± 575 | 106 ± 57 | 990 ± 864 | 381 ± 174 |
| PROTO | 7877 ± 703 | 583 ± 282 | 511 ± 298 | 1174 ± 291 | 874 ± 66 | 1696 ± 595 | 696 ± 120 |
| PEX | 4953 ± 454 | 83 ± 21 | 674 ± 177 | 1436 ± 482 | 501 ± 31 | 2960 ± 119 | 8320 ± 4187 |

## 6.2 ENVIRONMENTS AND DATASETS

We evaluate PTGOOD and our baselines on a set of environment-dataset combinations that satisfy two criteria: (a) it must not be possible for current algorithms to learn an optimal policy during the offline pretraining phase, and (b) we must be able to surpass a random agent during offline pretraining. If criterion (a) is violated, there is no need for online fine-tuning. If criterion (b) is violated, then the offline pretraining phase is not useful, and training from scratch online would be unlikely to be beaten. We use datasets in the Halfcheetah and Hopper environments from the D4RL study. Additionally, we collect our own datasets from environments not represented in D4RL, including Ant, Humanoid, and the Walker task from the DeepMind Control Suite (DMC). The datasets that we collect follow the same dataset design principles of D4RL. See Appendix C for more details on our environments and datasets.

## 6.3 OtO RESULTS

For each environment-dataset combination, we first pretrain agents offline to convergence and then fine-tune online for 50k environment steps across five seeds. Every 1k environment steps, we collect undiscounted returns across 10 evaluation episodes. Reporting comparative results between RL algorithms is a complex problem (Patterson et al., 2023); therefore, we present results across various views and mediums. Table 2 shows the average ± one standard deviation of evaluation returns at the 50k online-steps mark with the highest returns bolded. We highlight in blue when the highest returns are statistically significantly different via a two-sided Welch's t-test. Figure 11 displays undiscounted evaluation return curves for all algorithms in all environment-dataset combinations across the 50k online fine-tuning steps. Figure 12 displays undiscounted evaluation return curves in all five training runs for the best and second-best performing algorithms in each environment-dataset combination.

First, we answer question (1) in the affirmative by highlighting that PTGOOD consistently provides the strongest performance across all environment-dataset combinations. Table 2 shows that PTGOOD provides the highest returns in 7/7 environment-dataset combinations, which are statistically significant in 5/7. Also, Figure 11 shows that PTGOOD is generally stable relative to other baselines (e.g., RND/DeRL in Halfcheetah (Random)). We also note that PTGOOD tends to avoid the premature policy convergence that other methods sometimes exhibit (e.g., DMC Walker (Random), DMC Walker (Medium Replay), and Hopper (Random) in Figure 12). See Appendix E for more analysis. Also, aside from higher returns after training has finished, PTGOOD often outperforms other baselines during the middle portions of fine-tuning (e.g., Halfcheetah (Random) and Ant (Medium Replay) in Figure 12).

Second, we address question (2). We note that the Naive method is a strong baseline across all environment-dataset combinations that we tested. Additionally, we highlight that the Naive baseline outperforms some guided-exploration baselines on occasion (e.g., RND/DeRL in Halfcheetah (Random) and UCB(T) in Ant (Medium Replay)). These results suggest that certain types of exploration are not universally helpful in OtO RL.

Third, we answer question (3) by observing Cal-QL results in Table 2 and training curves in Figure 11. We note that Cal-QL performs poorly consistently. This is unsurprising because Cal-QL's base algorithm encourages the learned policy to remain close to the behavior policy. Due to our

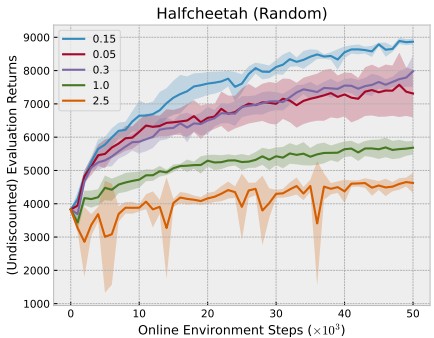 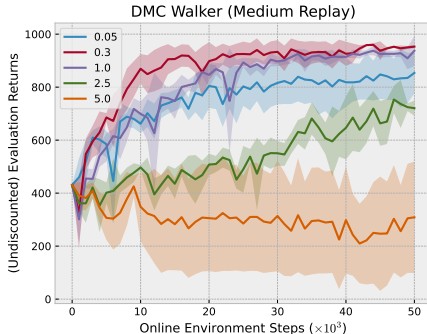

Figure 2: Average (bold line) ± one standard deviation (shaded area) of evaluation returns for the noise experiment in Halfcheetah (Random) (left) and DMC Walker (Medium Replay) (right).

environment-dataset selection criteria, the behavior policies are highly suboptimal, which makes conservatism and pessimism an unideal choice. Also, for the most part, our tuned baselines do not experience an initial performance collapse during online fine-tuning, as seen in Figure 11. This is in contrast to prior OtO work that focuses on bias correction due to the policy-constraint mechanisms in offline RL algorithms (e.g., Figure 2 in Nakamoto et al. (2023)). These results suggest that avoiding pessimistic and conservative methods may be a sensible choice in the OtO RL setting.

Finally, we note that neither UCB type is consistently better than the other. Additionally, in some environment-dataset combinations, either method is outperformed by the Naive baseline (e.g., in Halfcheetah (Random) for UCB(Q) and Ant (Medium Replay) for UCB(T)). This evidence, when combined with our experiment in §5.2, suggests that further research in multi-ensemble UCB exploration could prove fruitful.

## 6.4 Planning Noise

Key to our algorithm is exploring both unknown and high-reward regions of the state-action space. Instead of targeting high-reward state-action pairs with a Q-function value estimate, PTGOOD remains "close" to the improving policy by adding a small amount of noise to actions during the planning process. Using noise instead of explicit value estimation has computational benefits (see Appendix B) and does not rely on values that may be overestimated due to distributional shift (Fujimoto et al., 2018; 2019).

The meanings of "far" and "close" in the context of action selection are likely to be environment-dependent. As such, we perform a sweep over $\epsilon$ values in various environment-dataset combinations. Figure 2 shows the average ± one standard deviation of undiscounted evaluation returns for Halfcheetah (Random) and DMC Walker (Medium Replay) for various noise levels. We note that there is an optimal noise hyperparameter in either environment. If $\epsilon$ is too small, evaluation returns degrade slightly due to the reduced exploration. Also, if $\epsilon$ grows too large, PTGOOD's exploration strays too far from the improving policy and may become close to random exploration, which produces significantly reduced evaluation returns.

## 7 Conclusion

In this work, we introduced PTGOOD, a complement for model-based RL algorithms for exploration in the OtO setting. PTGOOD uses an estimate of the behavior policy's occupancy measure within a non-myopic planner to target high-reward state-action pairs unrepresented in the offline dataset. Also, we examined major online RL exploration paradigms, identified their compatibility issues with the OtO setting, and ultimately produced several strong baselines. We demonstrated that PTGOOD consistently provides the highest returns and avoids suboptimal policy convergence across our benchmark environments. PTGOOD could be improved further with adaptive noise in the planning process, which could account for state-dependent exploration noise or action-space characteristics (e.g., different joint types in musculoskeletal control).

## 8 REPRODUCIBILITY STATEMENT

We provide data from our OtO evaluation for each method tested. Also, we provide full visibility into baseline tuning with Figures 3-7. In addition, we open-source our code for each method, except for Cal-QL, which the original authors provide. We also provide details on architecture and hyperparameter settings in Appendix D.

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

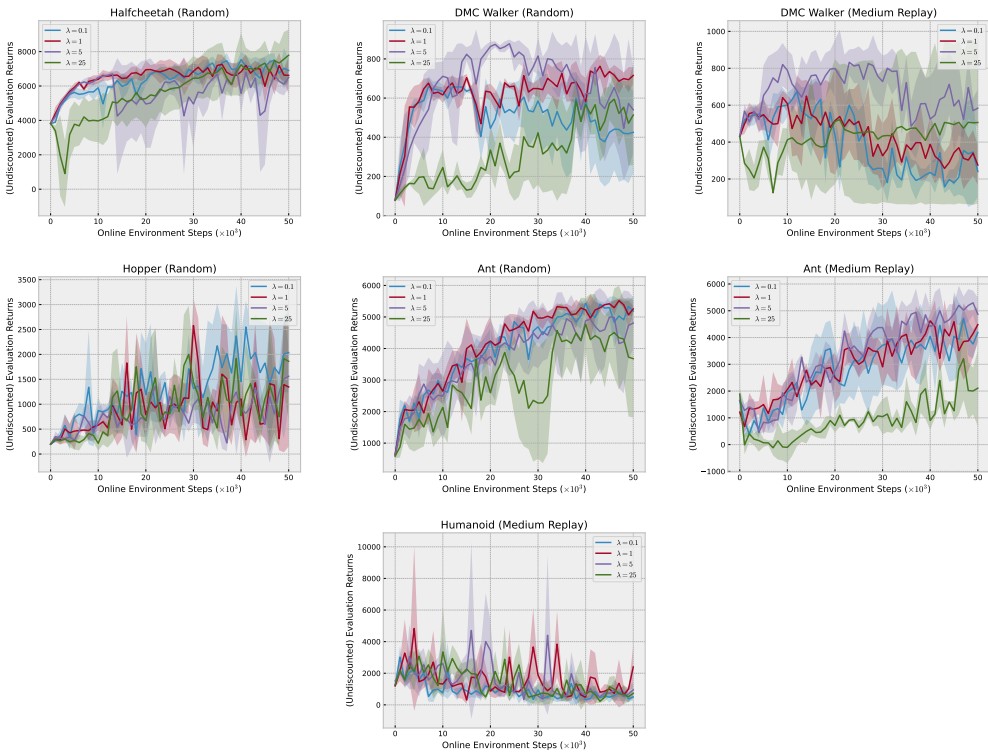

Figure 3: Undiscounted evaluation returns for RND/DeRL hyperparameter tuning.

# A    BASELINES

We use $\lambda$ as a generic weighting hyperparameter. For RND/DeRL (Figure 3), it weights intrinsic rewards at timestep $t$: $r_t = r_t^e + \lambda r_t^i$, and we scan $\lambda \in \{0.1, 5, 10, 25\}$. For UCB(Q) (Figure 4), it weights the impact of uncertainty on action selection: $Q_{mean}(\cdot) + \lambda Q_{std}(\cdot)$, and we scan $\lambda \in \{1, 10, 50\}$. For UCB(T) (Figure 5), it weights the impact of uncertainty on action selection: $Q(\cdot) + \lambda T_{std}(\cdot)$, and we scan $\lambda \in \{1, 10, 50\}$. For Cal-QL (Figure 6), it weights the Min Q-weight, which we found to be particularly impactful based on the hyperparameter sweeps found here: `https://wandb.ai/ygx/JaxCQL--jax_cql_gym_sweep_3`. In addition, we performed a sweep over the number of RL updates per environment step (Figure 7), called "UTD" in the Cal-QL paper. For Min Q-Weight, we scan $\lambda \in \{0.1, 1, 5, 25\}$, and for UTD we scan $\lambda \in \{1, 10, 20\}$.

For each hyperparameter setting, we run three seeds. Each plot shows the average (bold line) $\pm$ one standard deviation (shaded area). For the final results we present in the paper, we select the best performing hyperparameter setting for each algorithm on a per-environment basis and run two additional seeds.

# B    COMPUTE COST COMPARISON

We compare the wall-clock time of a PTGOOD planning process that uses only additive random noise (Noise) and one that uses additive random noise **and** computes Q-values (Q-values). We evaluate these two variations over three depths and widths (reported as width / depth): 100000 / 1, 50 / 3, 10 / 5. Specifically, we run each planning procedure for 10k environment steps five times and reports the average wall-clock time in seconds in Figure 8. We highlight that as soon as planning becomes non-myopic, using only noise provides significant gains in compute time.

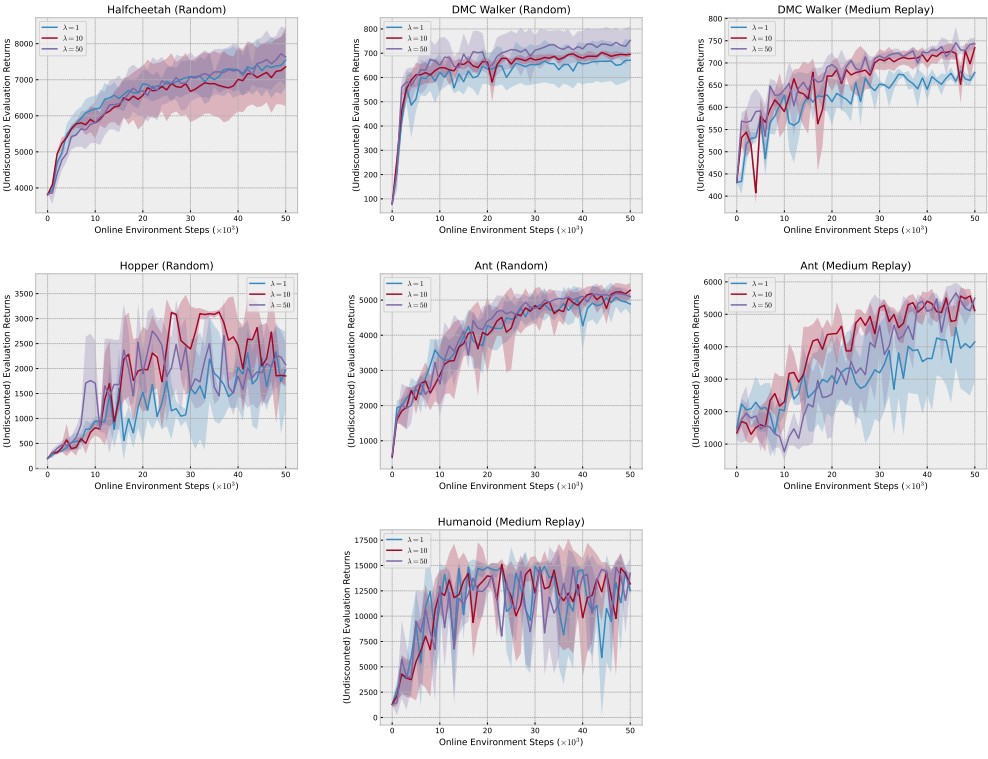

Figure 4: Undiscounted evaluation returns for UCB(Q) hyperparameter tuning.

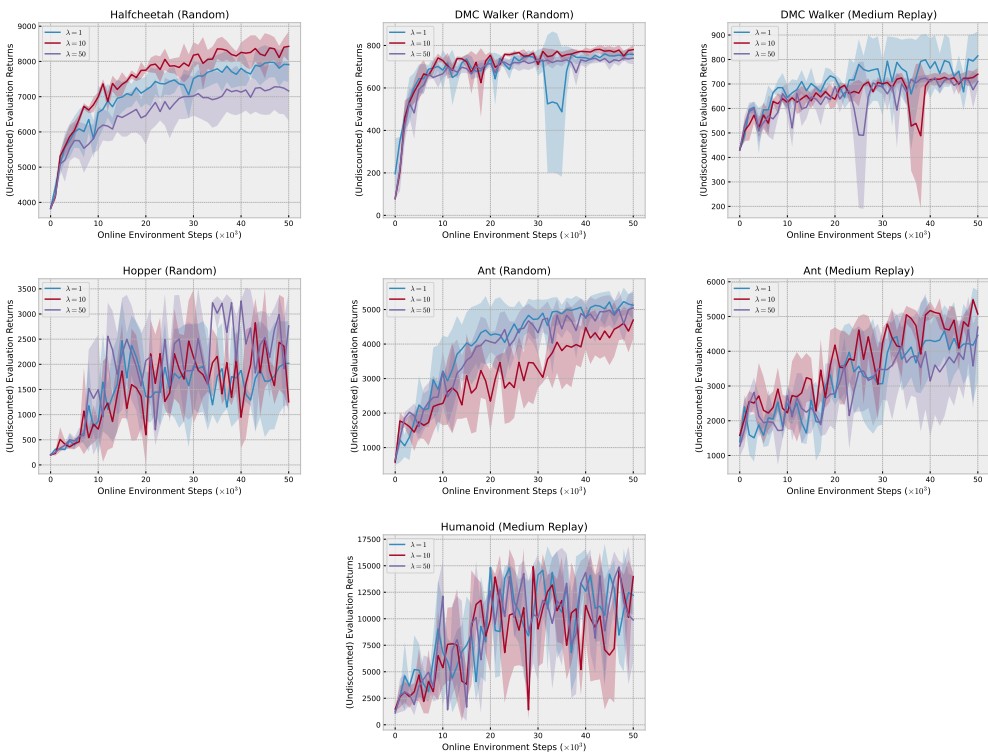

Figure 5: Undiscounted evaluation returns for UCB(T) hyperparameter tuning.

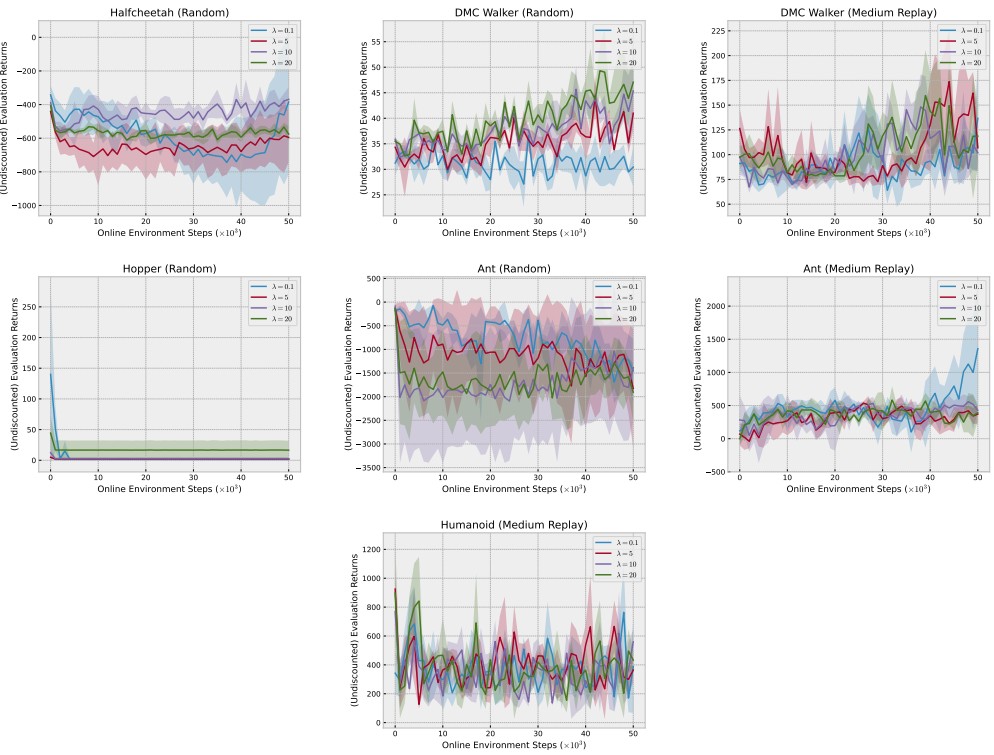

Figure 6: Undiscounted evaluation returns for Cal-QL (Min Q-Weight) hyperparameter tuning.

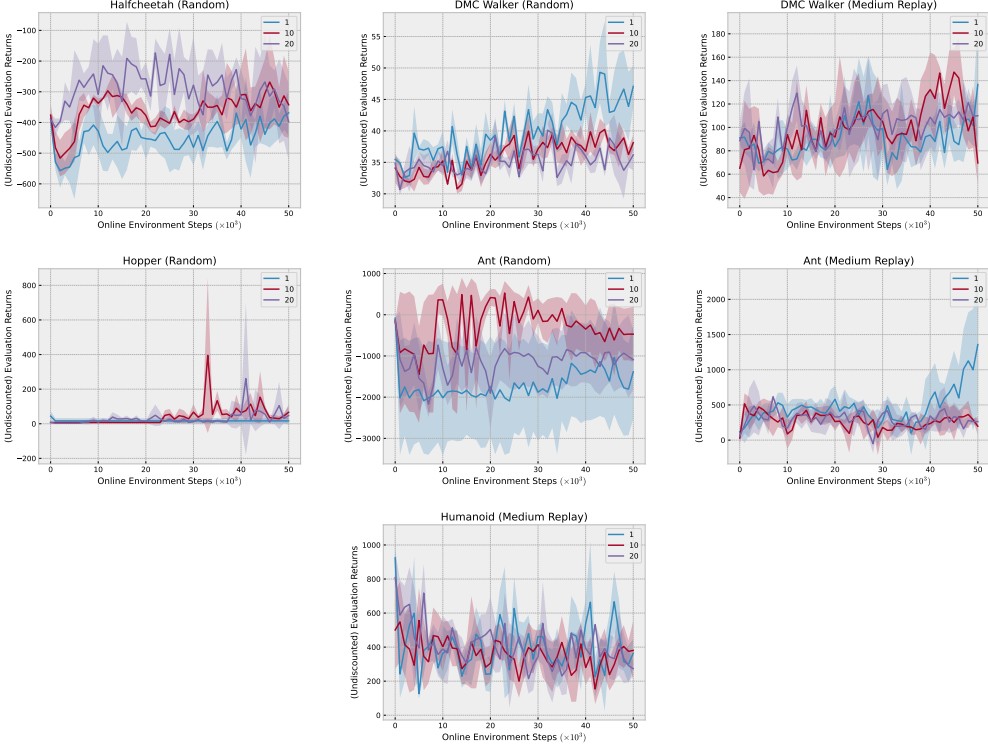

Figure 7: Undiscounted evaluation returns for Cal-QL (UTD) hyperparameter tuning.

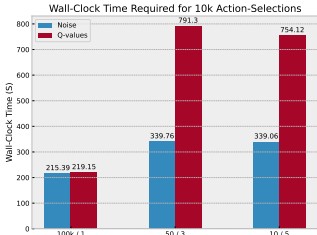

Figure 8: Wall-clock time (y-axis) comparison between noise-only planning (blue) and planning with Q-values (red) for three different width / depth combinations (x-axis).

## C  ENVIRONMENTS AND DATASETS

From the D4RL (Fu et al., 2020) dataset we use Halfcheetah (Random) and Hopper (Medium Replay). We collect our own datasets in the Walk task in Walker from DMC, the Walk task in Humanoid from the original MBPO (Janner et al., 2019) study, and the Walk task in the Ant environment from the original MBPO study. All (Random) datasets were collected with a policy that selects actions uniformly at random. All (Medium Replay) datasets were collected by saving the replay buffer of an MBPO+SAC agent trained purely online until "medium" performance. The medium performance is defined as generating evaluation returns of 400, 3000, and 6000 for DMC Walker, Ant, and Humanoid, respectively.

## D  ARCHITECTURE AND HYPERPARAMETERS

The MBPO+SAC agents use an ensemble of seven MLP dynamics models that parameterize Gaussians. In Humanoid environments, the MLPs are four layers with 800 hidden units each. In the Ant environments, the MLPs are four layers with 400 hidden units each. In all other environments, the MLPs are four layers with 300 hidden units each. All MLPs use *elu* activations. We train and perform inference in the same way as the original MBPO paper (see Table 1 in (Janner et al., 2019)). For any differences in hyperparameters, see Table 3. For environments with early-termination conditions, we zero out the rate value in states within the planning process that would terminate the episode to avoid incentivizing the agent to explore these paths.

Also, the MBPO+SAC agents use MLP actor and critic networks. In Humanoid and Ant environments, the MLPs are three layers with 512 hidden units each. In all other environments, the MLPs are three layers with 256 hidden units each. All MLPs use *elu* activations and the critic networks use layer norm operations. At each training step, data are sampled from the offline dataset, dataset of online interactions, and the model-generated synthetic transitions in equal parts.

The CEB encoder and decoder networks are both three-layer MLPs with 256, 128, and 64 hidden units and *elu* activations. The learned marginal is a Gaussian mixture model with 32 components.

All networks were trained with the Adam optimizer.The dynamics models used a learning rate of 1e-3 and a weight decay of 1e-5. The critic networks and learnable alpha were trained with a learning rate of 3e-4, while the actor networks used a learning rate of 1e-4. The target critic networks used a tau of 5e-3 with an update frequency of every other step.

For Cal-QL, we used the code and default architecture settings provided by the authors here: https://github.com/nakamotoo/Cal-QL.

UCB(Q) and UCB(T) both used seven ensemble members for their respective uncertainty computations.

RND/DeRL fine-tunes its RND predictor at the same frequency as its base agent updates its ensemble of world models (shown in Table 3).

Table 3: Hyperparameters used for PTGOOD and base MBPO+SAC agent.

| Environment-dataset | $\epsilon$ | $w$ | $d$ | imagination horizon | world model train freq | imagination freq |
|---|---|---|---|---|---|---|
| Halfcheetah (R) | 0.15 | 5 | 10 | 5 | 1000 | 1000 |
| DMC Walker (R) | 0.3 | 5 | 10 | 5 | 1000 | 1000 |
| Hopper (R) | 0.1 | 50 | 3 | 3 | 1000 | 1000 |
| Ant (R) | 0.025 | 50 | 3 | 3 | 250 | 250 |
| DMC Walker (MR) | 0.3 | 5 | 10 | 5 | 1000 | 1000 |
| Ant (MR) | 0.025 | 10 | 5 | 5 | 250 | 250 |
| Humanoid (MR) | 0.005 | 50 | 3 | 3 | 250 | 250 |

## D.1 THE CONDITIONAL ENTROPY BOTTLENECK

The Conditional Entropy Bottleneck (CEB) (Fischer, 2020) is an information-theoretic method for learning a representation $Z$ of input data $X$ useful for predicting target data $Y$. CEB's simplest formulation is to learn a $Z$ that minimizes $\beta I(X; Z|Y) - I(Z; Y)$, where $\beta$ is a weighting hyperparameter and $I(\cdot)$ denotes mutual information. Intuitively, CEB learns a representation that minimizes the extra information $Z$ captures about $X$ when $Y$ is known and maximizes the information $Z$ captures about $Y$. This form treats $X$ and $Y$ asymmetrically. Instead, the bidirectional CEB objective uses two separate representations $Z_X$ and $Z_Y$ for $X$ and $Y$, respectively:

$$\text{CEB}_{\text{bidir}} \triangleq \min - H(Z_X|X) + H(Z_X|Y) + H(Y|Z_X) \\ - H(Z_Y|Y) + H(Z_Y|X) + H(X|Z_Y), \tag{2}$$

where $H(\cdot)$ and $H(\cdot|\cdot)$ are entropy and conditional entropy, respectively. We can form Equation 2 as a self-supervised objective via a noise function $X' = f(X, U)$ with noise variable $U$, and treating the noised data $X'$ as the target $Y$. Additionally, Fischer (2020) show that we can place variational bounds on Equation 2 using a sampling distribution encoder $e(z_X|x)$, and variational approximations of the backwards encoder $b(z_{X'}|x')$, classifier $c(x'|z_X)$, and decoder $d(x|z_{X'})$ distributions. At convergence, we learn a unified representation that is consistent with both $z_X$ and $z_{X'}$ by applying the CEB objective in both directions with the original and noised data:

$$\min \langle \log e(z_X|x) \rangle - \langle \log b(z_X|x') \rangle - \langle \log c(x'|z_X) \rangle \\ + \langle \log b(z_{X'}|x') \rangle - \langle \log e(z_{X'}|x) \rangle - \langle \log d(x|z_{X'}) \rangle, \tag{3}$$

where each $\langle \cdot \rangle$ denotes the expectation over the joint distribution $p(x, x', u, z_X, z_{X'}) = p(x)p(u)p(x'|f(x, u))e(z_X|x)b(z_{X'}|x')$. We refer the reader to the original CEB paper for more details. Fischer (2020) show that we do not need to learn parameters for $c(\cdot)$ in Equation 3 because $c(x'|z_X) \propto b(z_X|x')p(z_{X'})$, which can be simplified further by marginalizing $p(z_{X'})$ over a minibatch of size $K$. The same can be done for $d(\cdot)$ using $e(\cdot)$. Altogether, this forms the contrastive "CatGen" formulation with the following upper bound:

$$\text{CEB}_{\text{denoise}} \leq \min_{e(\cdot), b(\cdot)} \mathbb{E} \left[ \mathbb{E}_{z_X \sim e(z_X|x)} [\beta \log \frac{e(z_X|x)}{b(z_X|x')} - \log \frac{b(z_X|x')}{\frac{1}{K} \sum_{i=1}^{K} b(z_X|x'_i)}] \right. \\ \left. + \mathbb{E}_{z_{X'} \sim b(z_{X'}|x')} [\beta \log \frac{b(z_{X'}|x')}{e(z_{X'}|x)} - \log \frac{e(z_{X'}|x)}{\frac{1}{K} \sum_{i=1}^{K} e(z_{X'}|x_i)}] \right] \tag{4}$$

where the outer expectation is over the joint distribution $x, x' \sim p(x, x', u, z_X, z_{X'})$.

## E SUBOPTIMAL CONVERGENCE

We highlight that many of our baselines' policies converge prematurely to suboptimal returns in both DMC Walker datasets. To help explain the phenomenon and describe how PTGOOD avoids this issue, we examine several metrics throughout online fine-tuning. Specifically, for UCB-style baselines, we examine ensemble disagreement and policy entropy. UCB-style methods sample the policy to create the set of actions over which disagreement is evaluated. Therefore, both of these

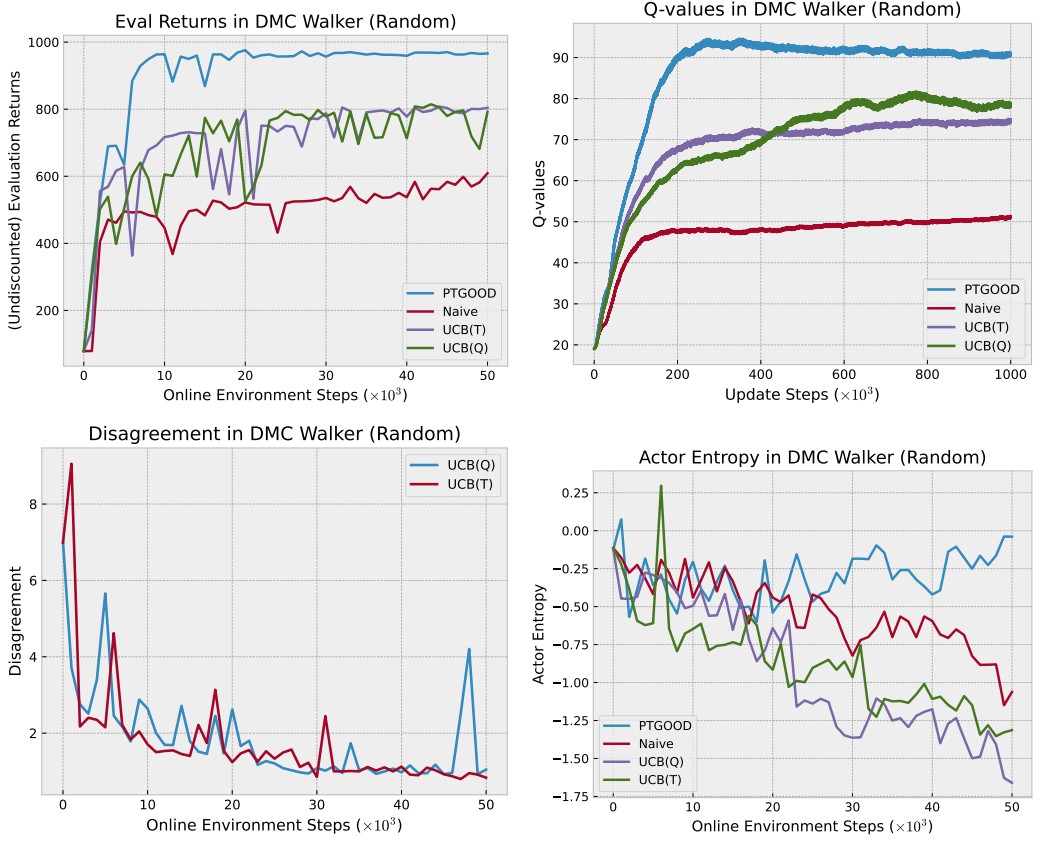

Figure 9: Metrics collected over 50k steps on online fine-tuning for the premature convergence experiment in DMC Walker (Random).

metrics drive exploration. For the other methods, such as Scratch and Naive, we examine only policy entropy. For these methods, the policies are sampled for action selection during online fine-tuning, and, therefore, its entropy is important for exploration. Both policy entropy and disagreement are captured during the evaluation episodes rolled out every 1k steps during online fine-tuning. We also capture average Q-values of each mini-batch used during agent training and evaluation returns. Finally, we collect all metrics except for disagreement for a PTGOOD agent. Figure 9 and Figure 10 show these metrics for DMC Walker (Random) and DMC Walker (Medium Replay), respectively.

We highlight that the disagreement metric for both UCB methods in both environment-dataset combinations starts relatively high but quickly collapses to a low number roughly around the time evaluation returns converge. Also, we note that the policy entropy of both UCB agents and the naive agent shows a consistent downward trend in both environment-dataset combinations. Such a reducing entropy will reduce the diversity in the action sets used for exploration in all three of these methods. In contrast, the PTGOOD agents' policy entropy remains relatively high throughout online fine-tuning.

Next, we show that the reduced exploration mentioned above causes the three baselines to miss exploring the same regions of the state-action space that PTGOOD explores. We demonstrate this by showing that the baselines' critics undervalue the state-action pairs collected by a higher-return PTGOOD agent and overvalue the state-action pairs that they themselves collect. If the baselines were to explore as well as PTGOOD, such erroneous Q-values would not exist. At the end of online fine-tuning, we collect 10 episodic trajectories of state-action pairs from each of the four agents. For their returns, see Figure 9 and Figure 10. Table 4 displays the average Q-values over the trajectories for each baseline in each environment-dataset combination.

# F    ADDITIONAL RESULTS

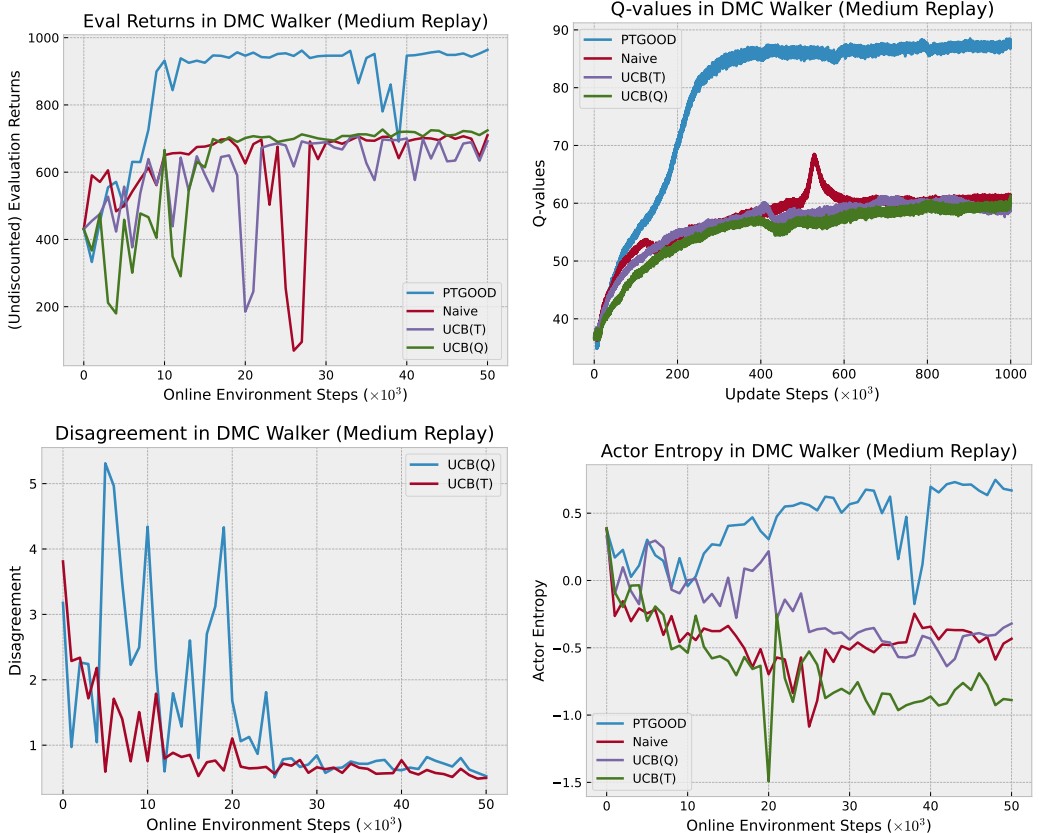

Figure 10: Metrics collected over 50k steps on online fine-tuning for the premature convergence experiment in DMC Walker (Medium Replay).

Table 4: Q-value over trajectory comparison for the premature convergence experiment.

| Dataset | Baseline | Q-value on PTGOOD trajectory | Q-value on own trajectory |
|---------|----------|------------------------------|---------------------------|
| MR | Naive | $51.1 \pm 4.3$ | $62.6 \pm 2.8$ |
| MR | UCB(T) | $48.6 \pm 3.1$ | $63.4 \pm 3.7$ |
| MR | UCB(Q) | $53.3 \pm 2.9$ | $64.1 \pm 3.8$ |
| R | Naive | $46.8 \pm 2.8$ | $54.1 \pm 4.3$ |
| R | UCB(T) | $71.6 \pm 3.9$ | $79.7 \pm 3.6$ |
| R | UCB(Q) | $68.7 \pm 2.4$ | $79.2 \pm 4.1$ |

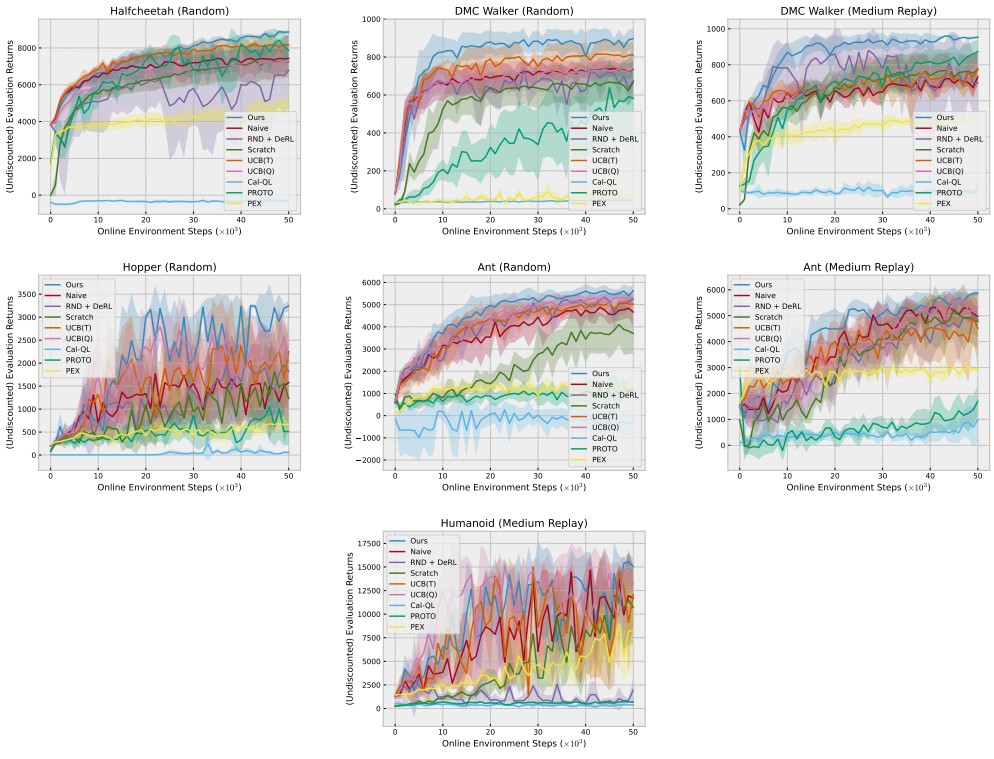

Figure 11: Undiscounted evaluation returns for all algorithms over the 50k online fine-tuning stage. Average (bold) ± one standard deviation (shaded area) displayed.

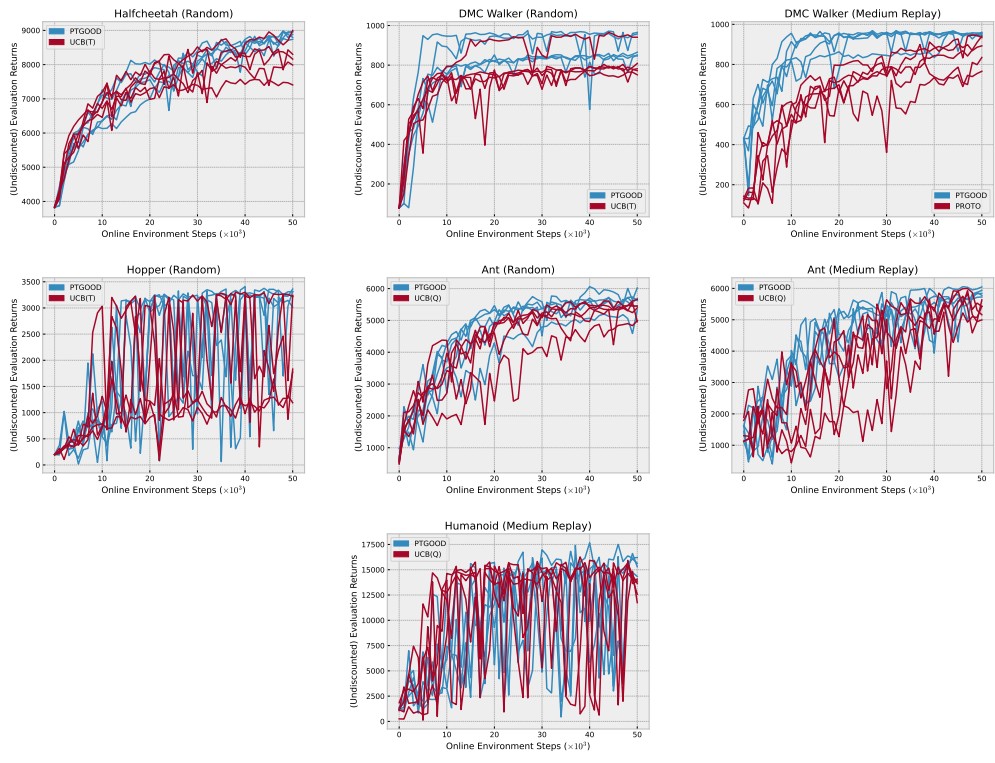

Figure 12: Undiscounted evaluation returns in all five training runs for the best and second-best performing algorithms over the 50k online fine-tuning stage.

