# OpenReview forum: "Planning to Go Out-of-Distribution in Offline-to-Online Reinforcement Learning"
_ICLR.cc/2024/Conference — Submitted to ICLR 2024_

### Official Review · Reviewer_Q4z6 · 2023-10-29

**Soundness:** 3 good
**Presentation:** 3 good
**Contribution:** 3 good
**Rating:** 6
**Confidence:** 3

**Summary:**

The paper introduces a new method for offline-to-online RL (oto RL), where the paper proposes that during the online fine-tuning stage, the policy perform exploration in a controlled manner and the exploration is determined by the visitation distribution of the current policy (or how out-of-distribution the state-action is). The paper proposes a way to measure the out-of-distribution-ness using conditional entropy bottleneck. Finally the paper compares the proposed algorithm with other baselines on a rage of offline-to-online benchmarks.

**Strengths:**

1. The paper proposes a natural way to address the oto RL problem: during the online fine-tuning stage, performing exploration to cover the unexplored region (thus out-of-distribution) seems a very reasonable strategy, and the paper suggests that one should also perform the exploration in a more careful manner (taking the return into account), which is a reasonable heuristic in practice.

2. The proposed algorithm is easy to understand and extensible (for a wide range of offline RL methods).

3. The empirical performance seems strong.

**Weaknesses:**

1. Although there seems no issue with the technical part of the paper, I do want to bring the attention to a recent paper: Reward-agnostic Fine-tuning: Provable Statistical Benefits of Hybrid Reinforcement Learning (https://arxiv.org/abs/2305.10282). I believe this paper, from the theory perspective, proposes the same intuition as the current paper: after running model-based offline RL on the offline dataset, one could use the model to estimate the occupancy measure of the offline policy, and thus have the knowledge of the uncovered directions from the offline data, and during online fine-tuning, one could use exploration to collect the data in the remaining directions. To me the current paper shares a lot of intuition with this earlier paper, minus many empirical considerations, which are also good contributions.

2. Other than CEB, there might be many other methods for measuring the out-of-distributioness. It would improve the paper if there are more ablations.

3. The experiment section seems to miss of of the benchmarks that are tested in cal-QL.

4. Minor point: in the related work section, two cited paper (Nair et al., 2020 and Song et al., 2023) seem to be more general than just using expert demonstration. I believe they also use the more general offline data (the same as the ot2 setting).

**Questions:**

See above.

---

> ### Author Response · Authors · 2023-11-20
> **Response to Reviewer Q4z6**
>
> We thank Reviewer Q4z6 for their time and comments. We are glad they found our work to be easy to understand and well-supported with strong empirical evidence. Please find our responses to your questions below, and we kindly ask whether Reviewer Q4z6 would be willing to increase their score or let us know if further clarifications are needed.
>
>
> **Suggested theory paper in tabular MDPs**
>
>
> We thank Reviewer Q4z6 for pointing us to this interesting reference. We were not aware of this work, but we have now included it in our Related Work section.
>
>
> **Other representation learning methods**
>
>
> In principle, representation learning methods other than the CEB can be used within PTGOOD. However, as we highlight in Section 4.2, CEB-learned representations have been shown to work well in out-of-distribution detection tasks, which is a property that is useful for PTGOOD’s purposes.
>
>
> **Request to include environments from the Cal-QL study**
>
>
> We specifically avoid using datasets that contain demonstrations of expert or hand-crafted policies (i.e., policies that successfully complete the task in the case of binary success tasks like Kitchen or optimal policies like in HalfCheetah Expert.) We motivate this setting in our introduction and reiterate it in our Related Work section. Below, we will go through each of the datasets used in the Cal-QL paper’s main results (Figure 6), and highlight that each of these datasets contains trajectories that violate this dataset selection criterion.
>
>
> All four of the AntMaze datasets used in the Cal-QL paper are collected by expert policies. The D4RL documentation (https://github.com/Farama-Foundation/d4rl/wiki/Tasks#antmaze) describes the trajectories as being generated by “commanding random goal locations in the maze and navigating the ant to them.” Also, the D4RL paper (Fu et al., 2021) states that the AntMaze datasets are generated by first “training a goal reaching policy.”
>
>
> All three of the Kitchen datasets are filled with trajectories that successfully complete the task. The documentation (https://github.com/Farama-Foundation/d4rl/wiki/Tasks#frankakitchen) states that the datasets “includes demonstrations of all 4 target subtasks being completed” and the D4RL paper describes the Kitchen datasets as coming from “human demonstrations” that are either “performing all of the desired tasks” or “performing subtasks.”
>
>
> The Pen, Door, and Relocate datasets are also filled with trajectories that successfully complete the task. The Cal-QL paper describes these datasets with: “Each of these tasks only provides a narrow offline dataset 	consisting of 25 demonstrations collected via human teleoperation and additional trajectories collected by a BC policy.” Additionally, the D4RL paper describes these datasets with: “This domain was selected to measure the effect of a narrow expert data distributions and human demonstrations.”
>
>
> The Cal-QL authors do not provide the dataset used in the Visual Manipulation environment, but they do refer to its datasets as containing “narrow” behavior, which is verbiage often used to describe datasets collected by expert policies. These datasets appear to come from Ebert et al., (2021), who describe the datasets as being collected via teleoperation specifically for imitation learning purposes, which suggests that the trajectories within the dataset successfully complete the tasks.

---

### Official Review · Reviewer_3FpJ · 2023-10-30

**Soundness:** 2 fair
**Presentation:** 2 fair
**Contribution:** 2 fair
**Rating:** 5
**Confidence:** 5

**Summary:**

This paper targets on the offline-to-online setting. Different with prior works, this paper frame offline-to-online setting as an exploration problem. For this reason, the authors study major online RL exploration paradigms and adpat them to work in this setting. This paper proposes an new method, named PTGOOD, which targets online exploration in relatively high-reward regions to encourage collect informative data. The authors show its performance in several tasks.

**Strengths:**

1. This paper is written well and easy to follow. The writing of the article is very clear.
2. The author gives a different perspective from previous work in offline-to-online setting, that is, using an exploratory approach to handle the switch from offline to online environments.

**Weaknesses:**

1. Although the authors tried to approach the offline-to-online problem from an exploratory perspective, they did not prove their claims through extensive experiments. For example, the experiments in Table 2 are too limited and only include 6 tasks. I suggest the authors provide additional experimental results in complete D4RL tasks to verify their claims.
2. In recent years, there has been rapid development in the field of offline-to-online RL, with numerous relevant works published. It is crucial for the authors to include and discuss these more recent works in the related works section, like AWAC[1], E2O[2], PROTO[3], SUNG[4] and PEX[5].
3. I have serious doubts about the reproduction of Cal-QL. Cal-QL does not seem to work at all in picture 11. This is very different from the results in the original paper. Why is this?

[1] Nair A, Gupta A, Dalal M, et al. Awac: Accelerating online reinforcement learning with offline datasets[J]. arXiv preprint arXiv:2006.09359, 2020.

[2] Zhao K, Ma Y, Liu J, et al. Ensemble-based Offline-to-Online Reinforcement Learning: From Pessimistic Learning to Optimistic Exploration[J]. arXiv preprint arXiv:2306.06871, 2023.

[3] Li J, Hu X, Xu H, et al. PROTO: Iterative Policy Regularized Offline-to-Online Reinforcement Learning[J]. arXiv preprint arXiv:2305.15669, 2023.

[4] Guo S, Sun Y, Hu J, et al. A Simple Unified Uncertainty-Guided Framework for Offline-to-Online Reinforcement Learning[J]. arXiv preprint arXiv:2306.07541, 2023.

[5] Zhang H, Xu W, Yu H. Policy Expansion for Bridging Offline-to-Online Reinforcement Learning[J]. arXiv preprint arXiv:2302.00935, 2023.

**Questions:**

1. The author claims that traditional exploration methods do not work, such as internal rewards and UCB. What will happen if naive exploration methods are used, such as epsilon exploration?
2. In the online stage, does the author use standard online RL algorithms, such as SAC and TD3, or does he use the online version of the offline learning algorithm?

-------

Thanks for the authors' explanation. I maintain my score since I believe this paper has a lot of room for improvement.

---

> ### Author Response · Authors · 2023-11-20
> **Response to Reviewer 3FpJ**
>
> We thank Reviewer 3FpJ for their time and comments. We are glad they found our paper to be well-written and easy to follow and that our work provides an interesting perspective on the offline-to-online problem. We have added additional baselines that Reviewer 3FpJ requested, which can be seen in our newly uploaded PDF. Please find our responses to your questions below, and we kindly ask whether Reviewer 3FpJ would be willing to increase their score or let us know if further clarifications are needed.
>
>
> **Request for more environments from D4RL**
>
>
> We specifically do not use all of the environment-dataset combinations in the D4RL suite because many of them violate our selection criteria (explained in section 6.2.) Many of the environment-dataset combinations in D4RL can be solved to (near-) optimality during the offline pretraining phase. Such environment-dataset combinations are uninteresting in the OtO setting and violate our criterion 1, which is defined in Section 6.2. On the other hand, some of the environment-dataset combinations provide such poor coverage of the MDP (e.g., the Random dataset in D4RL’s Walker2d) that offline pretraining provides virtually no benefit (violating criterion 2 which is defined in Section 6.2.) As a result, we collect several of our own datasets in environments that are not represented in D4RL, such as Ant, Humanoid, and DMC Walker, for a total of 7 environment-dataset combinations. These datasets comprise a portion of our work’s contribution to the research community, and we will release them to the public in our work’s code repository.
>
> **Request for more baselines**
>
>
> We have added two of the requested baselines (PROTO and PEX) to the main results in Table 2, Figure 11, and Figure 12. Two of the other requested baselines (E2O and SUNG) do not have code provided by the authors. The final baseline (AWAC) is outperformed by the two baselines we have already added, and therefore it would likely not provide additional information. We highlight that PTGOOD, our contribution, significantly outperforms both PROTO and PEX in all of our benchmarks.
>
>
> **Cal-QL performance**
>
>
> The environments used in the Cal-QL paper (Figure 6) contain trajectories of expert policies. This property is useful for Cal-QL since it contains a learning mechanism that keeps the learned policy close to the policy that collected the datasets (Equation 3.1 in the Cal-QL paper.) In contrast, we specifically avoid datasets that show expert behavior and instead only choose datasets collected by highly-suboptimal policies. We outline our dataset selection criteria in Section 6.2. As we explain in paragraph four of section 6.3, Cal-QL’s poor performance in these datasets should not be surprising, since it is actively keeping its learned policy close to the highly-suboptimal dataset-collection policy. The same should be true for any of the other algorithms that use the “conservative” and/or “pessimistic” mechanisms.
>
>
> Also, it is worth noting that the Cal-QL paper records agent returns over one million or more online fine-tuning steps. In contrast, our work records agent performance over only 50k timesteps, which represents a small fraction of the x-axis of the plots in Figure 6 of the Cal-QL paper.
>
>
> **Traditional exploration methods**
>
>
> We display in our experiment section (e.g., Table 2) that traditional exploration methods, such as intrinsic rewards and UCB exploration, work fine, but not as well as PTGOOD. Also, we perform hyperparameter tuning for each one of these baselines, as seen in Figures 3-7. We also show that following the policy without any exploration guidance (Naive) works fine but also significantly underperforms PTGOOD.
>
>
> **Algorithm used during online fine-tuning phase**
>
>
> All of our baselines (except for Cal-QL, PROTO, and PEX) and PTGOOD use model-based policy optimization (MBPO) and Soft-Actor critic (SAC), as we mention in the first paragraph of Section 6.1. Combining SAC with MBPO is standard practice and works well, as shown in the original MBPO paper (Janner et al., 2019).

---

### Official Review · Reviewer_Ftkq · 2023-10-31

**Soundness:** 3 good
**Presentation:** 1 poor
**Contribution:** 2 fair
**Rating:** 5
**Confidence:** 3

**Summary:**

This work focuses on offline-to-online (OtO) setting with limited budget online interactions. In particular, the proposed planning to go out of distribution (PTGOOD) treats this problem as an exploration problem and encourages the exploration on the dataset that is unlikely to be visited by the behavior policy. The experiments show that the proposed method can improve the learning performance comparing with previous methods.

**Strengths:**

1. This paper aims to solve an important problem in the OtO setting and the derived algorithm show the promising results in the DMC tasks
2. The exploration perspective is novel, which is in contrast with previous with regularization on the policy when exploring the out of distribution data.

**Weaknesses:**

1. The writing can be really hard to follow. The exploration approach is supposed to be used to motivate the proposed PTGOOD as in introduction. In section 4, many details of PTGOOD are referred to the following sections. The authors need to organize the paper in a different way
2. The core of the proposed method is to "target online exploration in relatively high-reward regions of the state-action space unlikely to be visited by the behavior policy". However, it is unclear what is "relatively high-reward regions", e.g., what is the criterial for choosing those regions.

**Questions:**

1. How does the learnt dynamics model $\hat{T}$ have impact on the PTGOOD planning procedure, e.g., the accuracy of the model vs. the performance of the PTGOOD
2. What is the explanation of the low variance in Figure 1 when choose small $\lambda$

---

> ### Author Response · Authors · 2023-11-20
> **Response to Reviewer Ftkq**
>
> We thank Reviewer Ftkq for their time and comments. We are glad they found our work to be novel. Please find our responses to your questions below, and we kindly ask whether Reviewer Ftkq would be willing to increase their score based on our responses or let us know if further clarifications are needed. Also, please see our updated PDF to see additional results we have added during the rebuttal period.
>
>
> **Question about finding relatively-high reward regions**
>
>
> We highlight in Section 6.4 that the relatively high-reward regions are found by keeping exploration “close” to the improving policy. Due to the agent’s learning process (via RL), it should visit increasingly higher-reward regions of the MDP’s state-action space naturally. As we show in the experiment in Section 6.4 and Figure 2, keeping the exploration process “close” to this improving policy is critical. This is shown by the large decrease in returns throughout training when the planning noise is too high.
>
>
> **Question about the relation between dynamics modeling accuracy and agent performance**
>
>
> This is an interesting question. Ideally, the learned dynamics model would be an oracle over the MDP’s entire state-action space (both in terms of reward- and transition-prediction.) However, our paper’s focus is on the limited-interaction setting. Therefore, we cannot afford to have the agent explore the MDP exhaustively, which would likely be required to achieve an oracle model. In our case, it is not so clear over which regions of the MDP’s state-action space we should measure the dynamics model’s accuracy and other performance metrics.
>
>
> Also, some recent works (e.g., Lambert et al., 2020; Wei et al., 2023), have shown that dynamics model prediction accuracy is not well correlated with agent performance. This is often called the “objective mismatch” between dynamics modeling and policy optimization of rewards. Because of this mismatch, it would probably be difficult to draw meaningful conclusions by observing modeling accuracy.
>
>
> **Explanation of the low variance in Figure 1**
>
>
> Could Reviewer Ftkq please clarify their question? In Figure 1, sometimes the variance across seeds is small when lambda is small, but sometimes the variance across seeds is large when lambda is small. The same is also true for relatively large values of lambda. So we are unsure what the reviewer is referring to specifically.

---

> > ### Comment · Reviewer_Ftkq · 2023-11-23
> > **Thank authors for the response**
> >
> > I thank the authors for the response.The quantification (e.g., error bound or characterization) on the key design details, such as "relatively high reward region" can be beneficial from the novelty perspective to this paper, especially the defined region is essential for the performance. Meanwhile the presentation of this work can be improved (see the original comments).  Overall I will keep my original rating.

---

### Author Response · Authors · 2023-11-20
**Thank you to the reviewers and new PDF**

We thank the reviewers for their time and valuable feedback on our paper. Please see the newly updated paper PDF above. We have added two additional baselines, as requested by Reviewer 3FpJ, and have added these results to Table 2, Figure 11, and Figure 12. We have also included additional citations as requested by Reviewer Q4z6.

---

### Meta-Review · Area_Chair_TSWa · 2023-12-10

**Metareview:**

Reviewers all scored the paper in the marginal reject range, citing issues with presentation, contribution, and lack of comparisons to recent methods. One found the exploration perspective novel but raised concerns about clarity of writing and defining key concepts like "high reward regions.". Another asked for more environments tested and discussion of related offline-to-online RL methods. The third thought the approach was reasonable but noted similarity to a recent theory paper and wanted more ablations. In response, the authors clarified details about their method, added experiments with more baselines as requested, and discussed differences from prior work. They explained their dataset criteria and why certain environments were excluded. However, reviewers maintained their original scores after the rebuttal.

Overall, the core weakness seems to be insufficient clarity, incremental contribution over recent work, and lack of thorough comparison to other offline-to-online RL methods. The authors made a reasonable effort to address feedback but not enough to significantly strengthen the paper.

**Justification For Why Not Higher Score:**

The paper is weakened by similar approaches, but additional responses did not significantly change the state.

**Justification For Why Not Lower Score:**

N/A

---

### Decision · Program_Chairs · 2024-01-16

Reject